# Simplifying glycan monitoring of complex antigens such as the SARS-CoV-2 spike to accelerate vaccine development

Janelle Sauvageau [1✉], Izel Koyuturk[2,3], Frank St. Michael[1], Denis Brochu[1], Marie-France Goneau[1], Ian Schoenhofen[1], Sylvie Perret[3], Alexandra Star [1], Anna Robotham[1], Arsalan Haqqani[1], John Kelly[1], Michel Gilbert[1] & Yves Durocher[2,3]

Glycosylation is a key quality attribute that must be closely monitored for protein therapeutics. Established assays such as HILIC-Fld of released glycans and LC-MS of glycopeptides work well for glycoproteins with a few glycosylation sites but are less amenable for those with multiple glycosylation sites, resulting in complex datasets that are time consuming to generate and difficult to analyze. As part of efforts to improve preparedness for future pandemics, researchers are currently assessing where time can be saved in the vaccine development and production process. In this context, we evaluated if neutral and acidic monosaccharides analysis via HPAEC-PAD could be used as a rapid and robust alternative to LC-MS and HILIC-Fld for monitoring glycosylation between protein production batches. Using glycoengineered spike proteins we show that the HPAEC-PAD monosaccharide assays could quickly and reproducibly detect both major and minor glycosylation differences between batches. Moreover, the monosaccharide results aligned well with those obtained by HILIC-Fld and LC-MS.

[1] Human Health Therapeutics Research Centre, National Research Council of Canada, 100 Sussex Dr., Ottawa, ON K1A 0R6, Canada. [2] Department of Biochemistry and Molecular Medicine, Faculty of Medicine, Université de Montréal, Montréal H3C 3J7, Canada. [3] Human Health Therapeutics Research Centre, National Research Council of Canada, 6100 Avenue Royalmount, Montréal, QC H4P 2R2, Canada. ✉email: Janelle.Sauvageau@nrc-cnrc.gc.ca

Since the beginning of the Severe Acute Respiratory Syndrome Coronavirus 2 (SARS-CoV-2) pandemic, a range of vaccine products such as messenger RNA vaccines, virus-like particles, and protein subunits have been developed for the purpose of protecting populations against this deadly virus[1]. Even though messenger RNA vaccines have received the most attention due to their quick development and proven efficacy, recombinant protein antigens represent the majority of candidates at the preclinical and clinical stage and many have recently been approved (e.g. Sanofi, Medicago, Novavax)[1]. In addition to being potential vaccine antigen candidates, these products are also exploited in antibody serology assays to detect anti-SARS-CoV-2 Immunoglobulin G in patient sera for the purpose of identifying infected individuals and evaluating vaccine efficacy[2].

The spike glycoprotein has up to 22 N-glycosylation sites per protomer (Supplementary Sequence 1) and each site can be modified with a range of glycans, resulting in a very complex glycosylation profile. Glycosylation varies from one site to the next (for example, high-mannose glycans are more abundant on certain sites) and is also dependent on the cell line in which the protein is produced[3]. Additionally, glycosylation can vary from batch-to-batch even when the same cell line and production conditions are followed. Importantly, major changes in glycosylation on the protein antigens can affect immunogenicity. For example, spike glycoprotein treated with endoglycosidase-H, resulting in the removal of high-mannose N-glycan and leaving an $N$-acetyl-D-glucosamine (GlcNAc) residue at each site, appears to elicit better protection from SARS-CoV-2 in mice than it's fully glycosylated counterpart[4]. Sialylation was also shown to affect the immunogenicity of the H5N1 influenza virus hemagglutinin antigens[5].

Despite this, the literature suggests that glycan analysis is not routinely performed on protein antigens and that glycosylation is assumed to be typical of that produced by the cells used to express them[6–8]. For example, it is known that the glycans on the surface of Human Embryonic Kidney (HEK-293) cells are mostly complex glycans containing sialic acid while insect cells express paucimannose glycans with no sialic acid. Also, glycosylation is impacted by the culture conditions and glycan structures may markedly vary between individual N-glycosylation sites. Without this information, it is not possible to assess the real impact of glycosylation on immunogenicity nor to compare it with data from other studies producing the same antigen.

Given the influence that glycans can have on critical product attributes in glycoprotein-based therapeutics such as stability and efficacy, it is important to have reliable assays for monitoring this post-translational modification. Regulatory agencies such as the U.S. Food and Drug Administration have issued guidelines related to glycan analysis for protein products[9,10]. There are well-established methods for assaying the relatively simple glycosylation found on biotherapeutic proteins such as monoclonal antibodies. For example, Hydrophilic Interaction Chromatography with fluorescence detection (HILIC-Fld) of N-glycans released with Peptide N-glycosidase F (PNGase F) works well in this context as it allows for the rapid identity of complete glycans when no co-elution occurs. The identity of glycan isomers by proteolytic digestion and glycopeptide analysis by Liquid Chromatography-Mass Spectrometry (LC-MS) is another powerful alternative and is especially useful to identify the location of the glycan and analyze glycan microheterogeneity on different sites, but requires a larger investment of time and resources to perform[11]. These methods are being used by the research community to characterize the extensive glycosylation on some SARS-CoV-2 spike protein constructs[12–20]. However, the HILIC-Fld profiles of spike protein N-glycans tend to be complex and so are challenging to interpret and quantify[14]. In addition, the huge spike glycopeptide data sets generated by LC-MS are not easy to use for comparing glycosylation between clones or batches. As such, there remains a need for a fast, reliable assay for monitoring the glycosylation of complex biotherapeutics like the SARS-CoV-2 spike glycoprotein, one that is easy to implement and fast. The latter trait is important as notable efforts are being invested currently in finding ways to shorten the development time for protein antigens so as to improve response speed in future pandemics[21].

High-Performance Anion Exchange Chromatography coupled with Pulsed Amperometric Detection (HPAEC-PAD) can be used to identify and quantitate the neutral and acidic monosaccharide content of glycoproteins[22–24]. Two separate analyses are performed: a neutral sugar analysis for the quantification of L-fucose (Fuc), N-acetyl-D-galactosamine (GalNAc), GlcNAc, D-galactose (Gal), D-glucose (Glc) and D-mannose (Man), and a sialic acid analysis for the quantification of $N$-acetylneuraminic acid (Neu5Ac) and $N$-glycolylneuraminic acid (Neu5Gc). These analyses are straightforward to perform, do not require labeling and the results are easy to analyze, there being only a handful of monosaccharide peaks to resolve and quantify. Other methods can be used for monosaccharide analysis, however, they require labeling or are less sensitive than HPAEC-PAD[25].

HPAEC-PAD was used in the past to analyze glycoproteins. It waned in popularity due to the development of techniques such as HILIC-FLD which are better suited for identifying the simpler glycan populations found on mAbs. While monosaccharide analysis does not provide glycan identity, it does give information about the average composition of monosaccharide units. In addition, the robustness of the method relative to other assays was questioned[26–28]. However, recent technological advances such as the introduction of disposable electrodes to eliminate fouling as well as the use of calibration curves to normalize the response differences between monosaccharides, have addressed most of these criticisms. Moreover, it has been shown that HPAEC-PAD can detect monosaccharides with unusual modifications such as mannose-6-phosphate which has been found on some glycoproteins[29]. It has to be determined yet if HPAEC-PAD can identify sugars with other modifications such as sulfation, a modification observed in low levels in the literature for the spike protein in HEK-293 cells[13,18]. Nevertheless, when the glycan profile is very complex, such as is the case with the SARS-CoV-2 spike glycoprotein, HPAEC-PAD could fill the gap left by established assays for a quick, reproducible, and quantitative monitoring assay.

To explore whether HPAEC-PAD could be used to quickly monitor glycans on complex antigens, two cohorts of spike proteins, one with major and the other with minor differences in glycosylation, were used for analysis. For the first set, five spike proteins with different N-glycosylation patterns were produced in engineered Chinese hamster ovary (CHO) cell lines (Table 1). A wild-type CHO phenotype (WT CHO[55E1]) containing N-glycans composed of Fuc, GlcNAc, Gal, Man, and Neu5Ac[30,31], was chosen as a control[32]. To assess the impact of changes in Fuc and Neu5Ac content, the spike protein was produced in three CHO knockout (KO) cell lines: F15 (for Fucosyl-transferase 8 (*FUT8*) KO, absence of core Fuc), S9 (for Sialyl-transferase 4 (*ST3GAL4*) KO, near abolition of 2,3-linked Neu5Ac on N-glycans) and dKO2 (for *FUT8* and *ST3GAL4* double KO). Finally, the spike protein was also expressed in WT CHO[55E1] supplemented with kifunensin, an inhibitor of mannosidase I, to produce only high-mannose N-glycans (mainly Man9 structures). The second set of spike proteins were produced from stable pools and were expected to display the minor differences in glycosylation that would be more likely to be encountered when comparing antigen production lots. Both spike protein cohorts were characterized using the three assays to determine if differences detected by HILIC-Fld and LC-MS were also

reflected in the simpler and faster HPAEC-PAD assays. The experimental plan is outlined in Fig. 1.

**Table 1 CHO cell lines used to produce spike glycovariants.**

| CHO cell line (spike lot number) | Genotype | Phenotype | Representative N-glycan structure |
|---|---|---|---|
| F15 (PRO1-468) | *FUT8* KO | -Fucosylation +α2,3 sialylation | |
| S9 (PRO1-469) | *ST3GAL4* KO | +Fucosylation -α2,3 sialylation | |
| dKO2 (PRO1-470) | *FUT8* KO and *ST3GAL4* KO | -Fucosylation -α2,3 sialylation | |
| WT CHO[55E1] (PRO1-471) | wt | +Fucosylation +α2,3 sialylation | |
| WT CHO[55E1] + Kifunensine (PRO1-472) | wt | +high-mannose structures -Fucosylation -α2,3 sialylation | |

Glycans are drawn using DrawGlycan-SNFG software[47]. Green circle: Man, blue square: GlcNAc, yellow circle: Gal, red triangle: Fuc, purple diamond: Neu5Ac.

Our findings show that the major glycosylation differences in the spikes produced in the engineered CHO cell lines identified by HILIC-Fld and LC-MS could also be detected by HPAEC-PAD. The monosaccharide assays could also detect minor variations in the three batches obtained from the same stable pool and this also correlated with the results from the HILIC-Fld and LC-MS analyses. Thus, neutral sugar and sialic acid analyses using HPAEC-PAD can be an effective and robust alternative to the established HILIC-Fld and LC-MS assays for lot-to-lot monitoring of glycosylation of the SARS-CoV-2 glycoprotein and likely for other complex proteins also.

## Results

### Glycosylation analysis of spike protein from engineered CHO cell lines

*Monosaccharide analysis by HPAEC-PAD.* The neutral and acidic monosaccharide content of the spike proteins with various glycosylation patterns was determined using two separate assays. For neutral monosaccharide analysis, three aliquots of each spike protein were hydrolyzed with trifluoroacetic acid (TFA), injected in triplicate on an HPAEC-PAD system, and eluted with a sodium hydroxide (NaOH) gradient (Fig. 2a, and Supplementary Table 1). The average and standard deviation of the three hydrolysates and injections ($n = 9$) are reported in mol of monosaccharide mol$^{-1}$ of protein (protomer: 143,230.6 Da)

The monosaccharides Fuc, Gal, Man, and GlcN were detected in the hydrolysate of the WT spike protein (PRO1-471). D-Glucosamine (GlcN) is the N-deacetylation product of GlcNAc resulting from TFA treatment[23]. These monosaccharides are the main constituents of complex N-glycans and their presence is

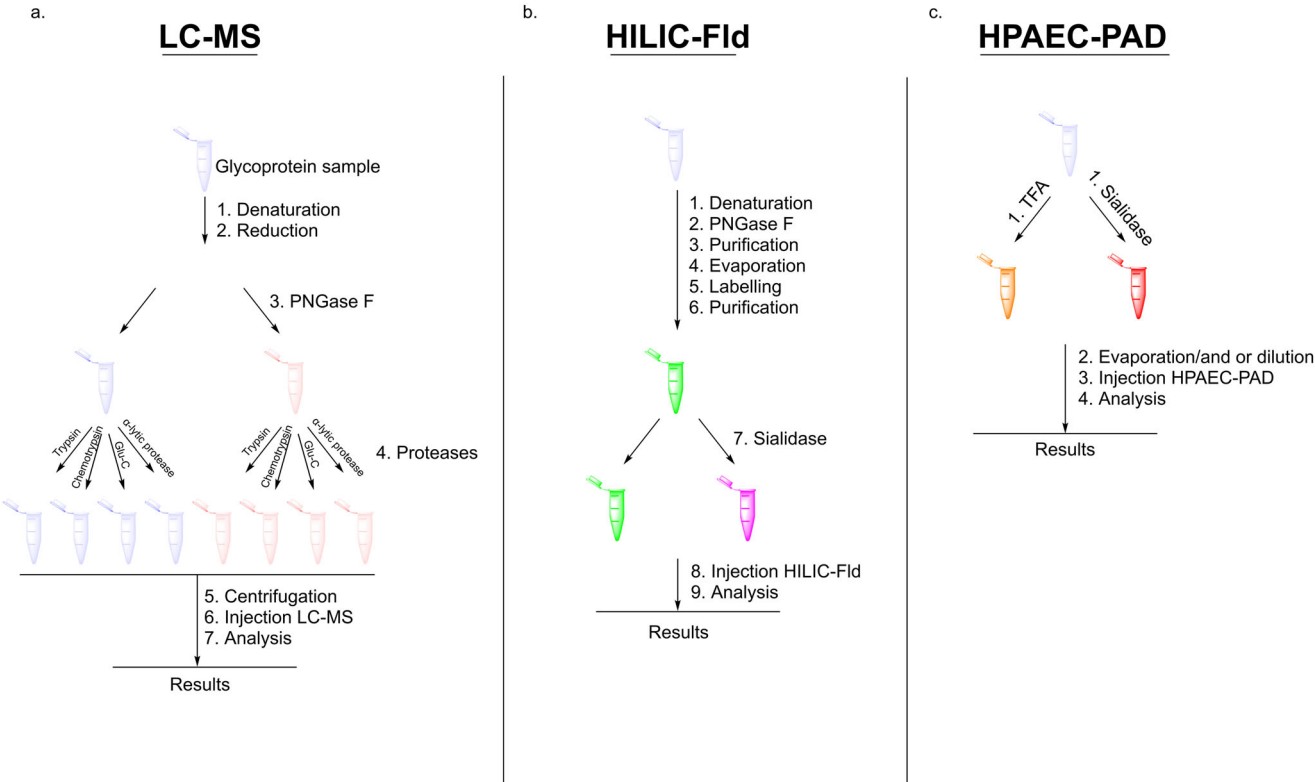

**Fig. 1 Comparison of the sample preparation required for LC-MS, HILIC-Fld, and HPAEC-PAD. a** Illustrates the seven required steps to obtain the results via LC-MS, encompassing denaturation, reduction, PNGase F and protease digestion, centrifugation, injection, and analyses. **b** The nine necessary steps to acquire the results are outlined, involving denaturation, PNGase F digestion, purification, evaporation, labeling, purification, sialidase treatment, injection, and analyses. Lastly, **c** highlights the four essential steps prior to obtaining the results via HPAEC-PAD, encompassing TFA or sialidase cleavage, evaporation, injection, and analyses.

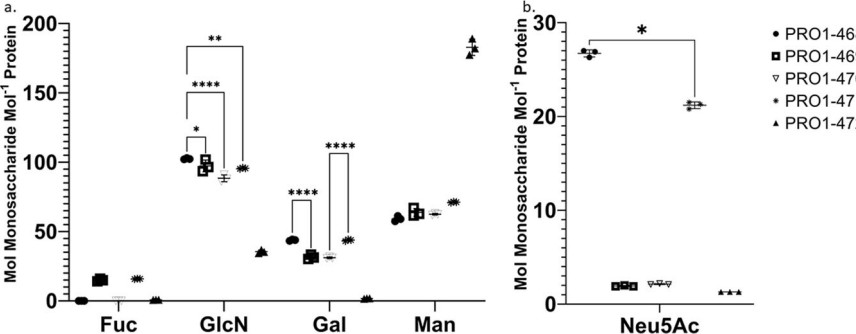

**Fig. 2 Monosaccharide analysis via HPAEC-PAD of glycovariants of the SARS-CoV-2 spike protein.** Results reported in mol monosaccharide mol$^{-1}$ monomer (protomer MW = 143,230.6 Da). Results are means of three triplicate injections and three triplicate reactions. Error bars are the standard deviation observed between the triplicate hydrolyses values. **a** Fuc, GlcN, Gal, and Man content obtained after TFA hydrolysis. The data were analyzed using two-way ANOVA, Tukey's multiple comparisons test, comparing the content of each monosaccharide type between variants. Not all comparisons shown. For full comparisons see Supplementary Table 2. **b** Neu5Ac content obtained after enzymatic hydrolysis. For a full comparison see Supplementary Table 3. $P$ values: <0.03 (*), <0.002 (**), and <0.0001(****).

consistent with the previously published data for the spike protein[12,13]. These same monosaccharides were observed in spike produced from the *ST3GAL4* KO (PRO1-469 and PRO1-470), although Gal abundance was approximately 10 mol/mol lower (i.e., 27–29% reduction) than what was observed for WT (PRO1-471). Fuc was not detected in spikes from either of the *FUT8* KO (PRO1-468 and PRO1-470) indicating that their N-glycans are entirely afucosylated. In addition, HPAEC-PAD analysis detected a clear difference in neutral monosaccharide composition between the kifunensin-spike protein (PRO1-472) and the other variants. Man and GlcN were by far the most abundant monosaccharides in the PRO1-472 hydrolysate and the Man content was 2.6 times higher than what was observed for WT spike protein (PRO1-471). This result was expected given that kifunensin, a mannosidase I inhibitor, promotes the expression of high-mannose glycans at the expense of complex and hybrid N-glycans. GalN, the product of deacetylation of GalNAc after TFA hydrolysis, was not detected in any of the samples. GalNAc is a component of O-linked glycans that are attached to the side chains of serines and threonines[33]. The lack of detectable GalN in these hydrolysates suggests that O-linked glycans are either entirely absent from these proteins or are present at levels too low to be detected by this assay.

Next, the acidic monosaccharide content (i.e., Neu5Ac and Neu5Gc) was evaluated by treating the samples with Sialidase A in triplicate and injecting the released sialic acids on an HPAEC-PAD system with a NaOH and a sodium acetate linear-gradient (Fig. 2b, and Supplementary Table 1). As a control, the WT CHO (PRO1-471) exhibited 21.2 mol Neu5Ac mol$^{-1}$ protein. PRO1-468 (*FUT8* KO) had a similar Neu5Ac content (26.7 mol Neu5Ac mol$^{-1}$ protein). Unsurprisingly, PRO1-469 and PRO1-470 produced in *ST3GAL4* KO cell lines contained little Neu5Ac (2.0 and 2.1 mol Neu5Ac/mol protein, respectively). In these KO cells, the main sialyltransferase (*ST3GAL4*) that transfers 2,3-linked Neu5Ac onto *N*-acetyllactosamine branches is absent although other sialyltransferases can possibly contribute to the low sialylation observed[34]. Similarly, Neu5Ac content is low (1.3 mol Neu5Ac mol$^{-1}$ protein) in PRO1-472 (WT CHO supplemented with kifunensin). These glycans do not have terminal Gal which prevents the addition of Neu5Ac[35]. Traces of Neu5Gc was also detected in most samples (Supplementary Table 1) but at very low levels (0.1 mol Neu5Gc mol$^{-1}$ protein) confirming that in CHO cells Neu5Ac hydroxylase is present but not expressed as reported previously by Wang et al[36].

*HILIC-Fld analysis of released N-glycans.* N-glycans were released from the spike proteins PRO1-468 to -472 with PNGase F, labeled with 2-aminobenzamide (2-AB), and separated on a HILIC-UPLC-Fld system (Fig. 3). The retention times were calibrated with a 2-AB labeled dextran ladder, converted to Glc units (GU) values and used to search the Glycobase database for glycan identities[37]. Usually, glycan retention times are correlated to their polarity, with larger and negatively charged glycans appearing at higher GU values. This is usually a straightforward analysis for glycoproteins with simple N-glycan profiles, such as mAbs[38]. However, all the spike proteins analyzed here (except for PRO1-472), yielded very complex glycan profiles, making it difficult to assign an identity to all but a handful of the major peaks. In contrast to the other HILIC traces, only high-mannose glycans (Man4-9) were detected in PRO1-472 with Man9 being the most abundant. This glycan profile is expected for a glycoprotein expressed in CHO cells in the presence of kifunensin.

Despite challenges in identifying the majority of glycans detected in the HILIC traces, some inferences about the glycosylation of the different spike glycovariants can be made (Fig. 3). For example, the HILIC chromatograms for PRO1-468 and PRO1-471 contain glycan signals with high GU values that are characteristic of sialylated glycans, while these peaks are absent from PRO1-469 and PRO1-470 (*ST3GAL4* KO) and PRO1-472. Furthermore, prominent peaks corresponding to the fucosylated biantennary glycans, FA2, FA2G1, and FA2G2, are detected in PRO1-469 and PRO1-471 while peaks of similar intensity corresponding to afucosylated biantennary glycans, A2, A2G1, and A2G2, are present in PRO1-468 and PRO1-470, implying that fucosylation is impaired or absent in the *FUT8* KO. Note that it was not possible to determine the degree of fucosylation from the HILIC-Fld chromatograms of the N-glycans as most of the peaks could not be identified.

Additionally, it is also possible to draw some conclusions about the relative levels of galactosylation in the spike protein glycovariants. In Fig. 4 the HILIC-Fld chromatogram for *ST3GAL4* KO PRO1-469 and WT PRO1-471 after treatment with sialidase show that for PRO1-471, the biantennary galactosylated FA2G2 peak is higher than the biantennary non-galactosylated FA2 peak. While for PRO1-469 (*ST3GAL4* KO), the opposite is observed with the intensity of FA2 being higher than the intensity of FA2G2. This result aligns with the data from the monosaccharide analysis (Fig. 2a and Supplementary Table 1) showing higher galactosylation for PRO1-471 than PRO1-469. A similar difference in galactosylation is observed for PRO1-470

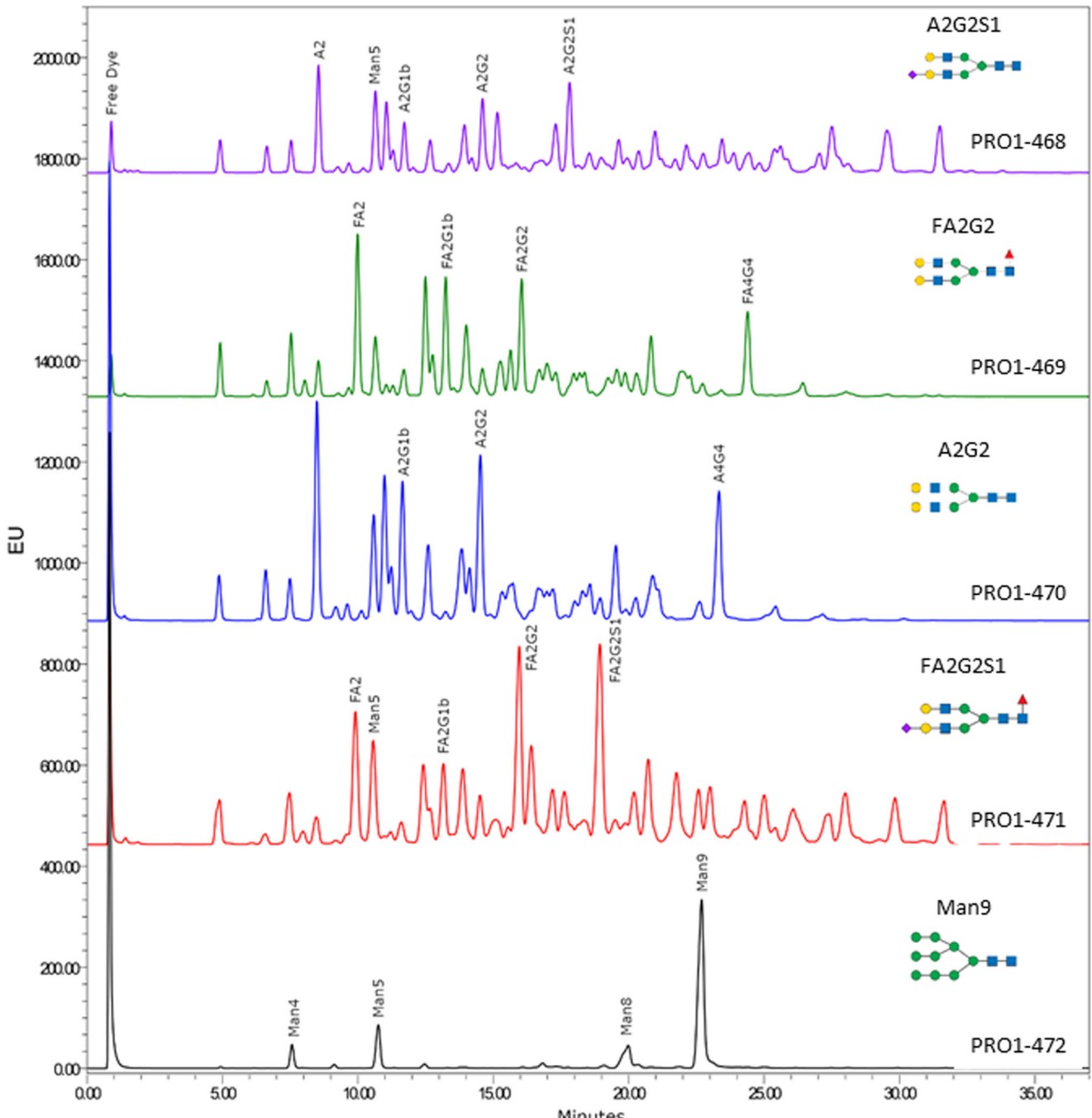

**Fig. 3 HILIC-UPLC-Fld chromatograms of PRO1-468 to PRO1-472 samples.** The chromatograms of PRO1-468 (purple), PRO1-469 (green), PRO1-470 (blue), PRO1-471 (red), and PRO1-472 (black) are represented with no offset on the x axis, but offsets on the y-axis to facilitate comparison. An expected representative structure is depicted on the right. To note, this representative structure indicates only a single version of many different glycoforms present. Glycans are depicted by the symbolic nomenclature for glycans using DrawGlycan-SNFG software[47]. Green circle: Man, blue square: GlcNAc, yellow circle: Gal, red triangle: Fuc, purple diamond: Neu5Ac.

and PRO1-468 in the monosaccharide analysis (Supplementary Fig. 1) and the relative abundance of biantennary glycan A2G2 vs A2 in the de-sialylated HILIC-Fld results. Therefore, it can be concluded that the level of galactosylation of spike protein N-glycans is lower in the *ST3GAL4* KO than in those expressed with active *ST3GAL4*.

*LC-MS glycopeptide analysis*. Next, the spike proteins were analyzed by bottom-up LC-MS to characterize and compare the glycosylation at each of the N-glycosylation sites. In keeping with

the approaches taken by other groups[12–15,19,39] a protocol using four proteases (trypsin, chymotrypsin, alpha-lytic protease, and glutamyl endopeptidase Glu-C) was employed to ensure that each N-glycosylation site could be isolated on its own and on a peptide that was compatible with LC-MS. Glycopeptides with 2 or more N-glycosylation sites were ignored in this analysis. Additionally, a portion of each spike protein digest was treated with PNGase F to remove N-glycans and analyzed by LC-MS/MS. The data acquired for these deglycosylated samples helped to confirm the peptide sequences and to eliminate false positives. GlycoPIQ, an

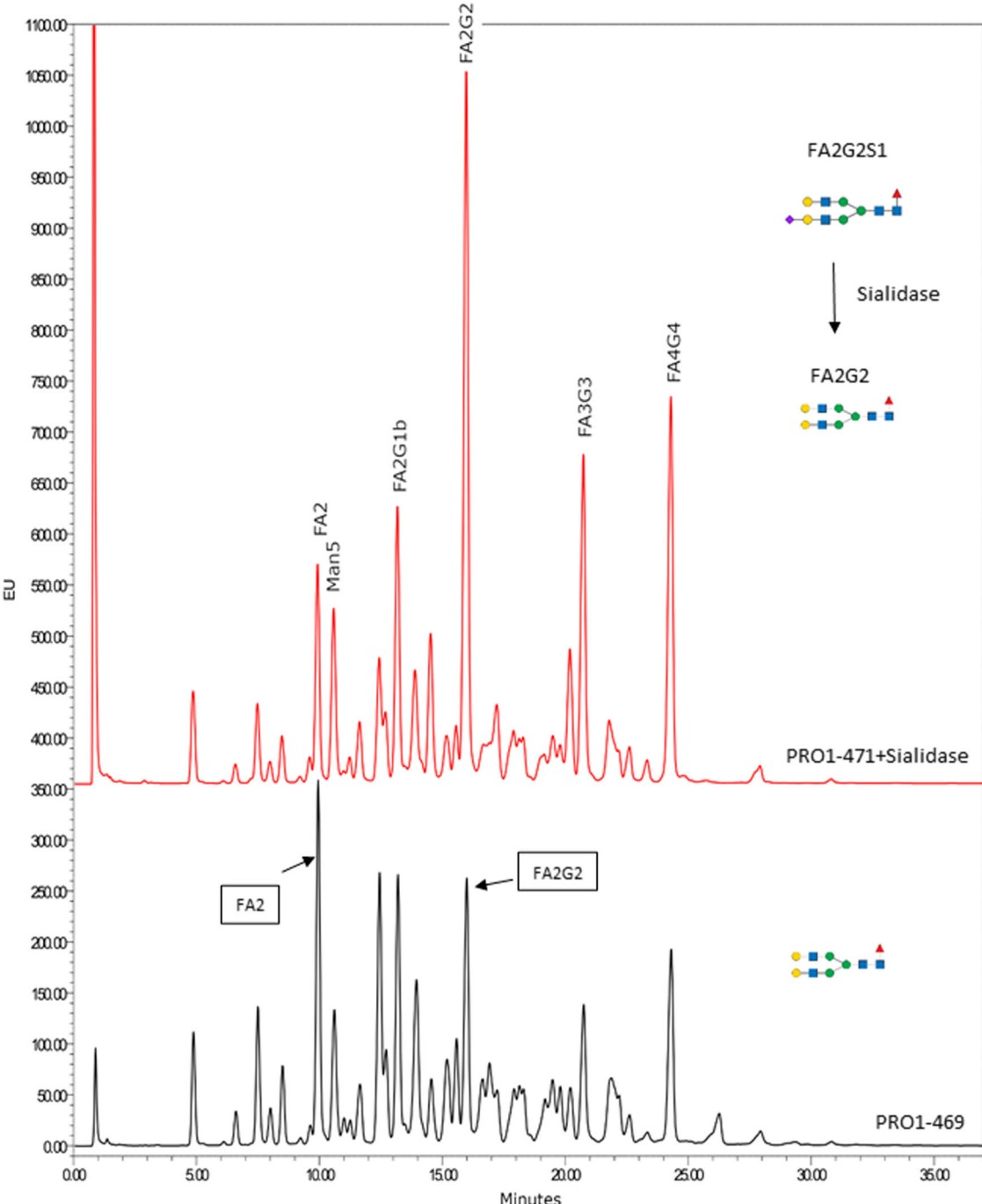

**Fig. 4 Comparison of HILIC-UPLC-Fld chromatograms with no offset on the *x* axis, but offsets on the *y* axis to facilitate comparison of PRO1-471 (red) after sialidase treatment and PRO1-469 (non-treated, black) with expected typical structure.** Glycans depicted as by the symbolic nomenclature for glycans using DrawGlycan-SNFG software[47]. Green circle: Man, blue square: GlcNAc, yellow circle: Gal, red triangle: Fuc, purple diamond: Neu5Ac.

in-house algorithm, was used to identify the glycans at each site and to determine their relative abundances (Supplementary Data 1).

Glycopeptides that matched the above criteria were found for 21 of the 22 N-glycosylation sites (Supplementary Table 4). Unfortunately, due to poor intensities and interference from other ions, neither peptides nor glycopeptides covering N-glycosylation site N1134 could be detected with confidence in all of the spike sample digests. The remaining 21 N-glycosylation sites were modified with a range of high-mannose, complex, and/or hybrid glycans of differing relative abundances. Some glycosylation sites were observed partially

unoccupied. A tabulation of the glycans observed on the 21 N-glycosylation sites is presented in Supplementary Table 5. Also, as an example, the glycopeptide categories identified by GlycoPIQ for 6 of the 21 N-glycosylation sites are presented in Fig. 5. The size of each segment in the pie charts is determined by the sum of the relative intensities of the glycopeptides assigned to that category.

The overall glycosylation patterns observed for the WT spike protein (PRO1-471) aligned well with the average glycosylation for recombinant spike proteins reported in other studies[12–14]. High-mannose glycans were predominant on-site N234 and to a lesser extent on N61 and N717. The remainder of the sites were

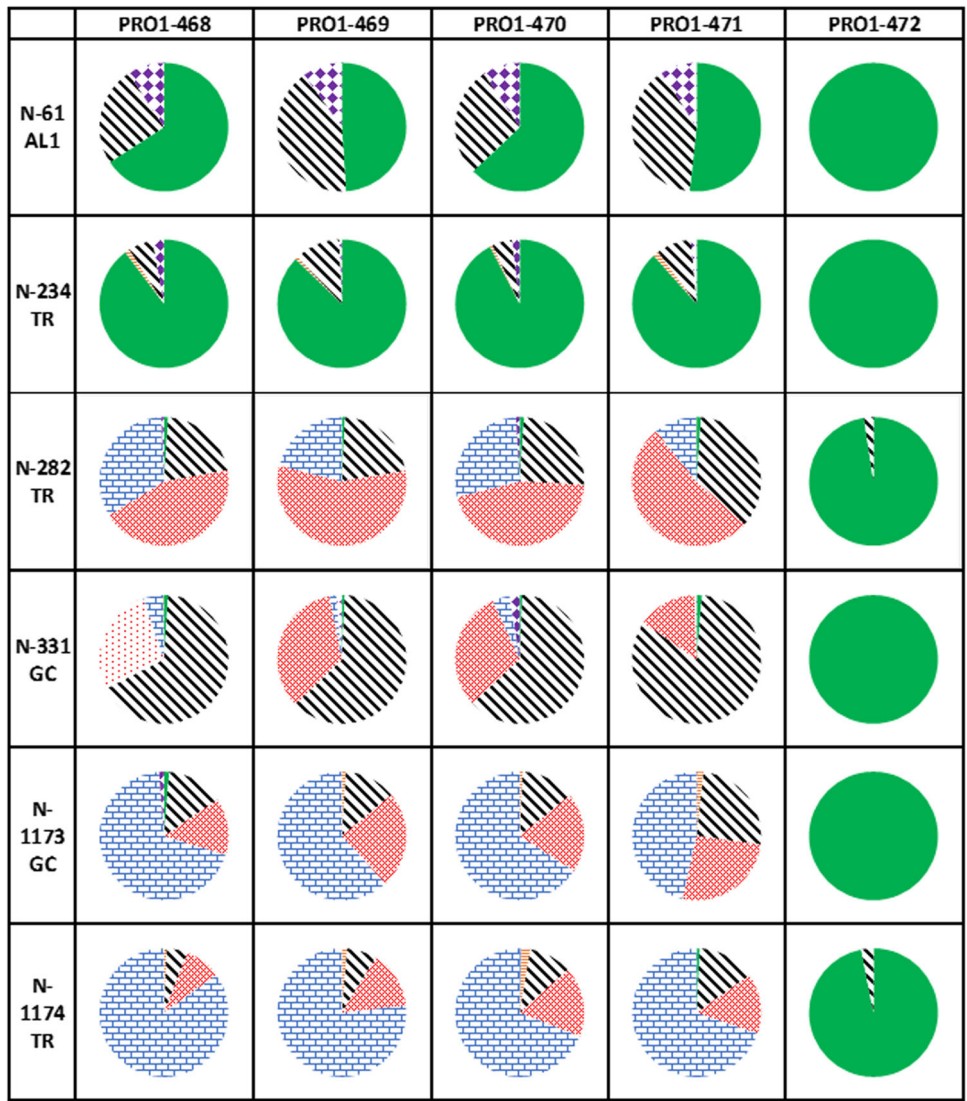

**Fig. 5 Example of glycan distribution per site.** Pie charts showing how glycans are distributed for six different sites. AL: α-lytic protease, GC: Glu-C, TR: trypsin. biantennary (black, wide diagonal stripes downward), triantennary (red, 30% dotted), tetra-antennary (blue, horizontal brick), hybrid (orange, narrow horizontal stripes), other glycans (includes glycans such as A1 and A1G, purple, solid diamond grid) and high-Man (green, solid fill).

occupied mostly with complex glycans. Interestingly, tri- and tetra-antennary glycans were more prevalent on sites nearer to the C-terminus (Supplementary Table 5). As expected, only trace amounts of sialylated glycopeptides were detected in the digests of the *ST3GAL4* KO spike proteins, PRO1-469 and PRO1-470. In contrast, most of the sites on both PRO1-468 and PRO1-471, with the exception of those dominated by high-mannose glycans, were occupied to some degree with sialylated glycans. Sialylation levels were greater on those sites with more multi-antennary glycans. Interestingly, PRO1-468 displayed a greater level of sialylation than PRO1-471. This is in keeping with the fact that PRO1-468 also expressed more of the larger, tetra-antennary glycans as was observed by HILIC-Fld and LC-MS (Supplementary Fig. 1 and Table 5).

Interestingly, the glycosylation differences between proteins observed at the individual site level by LC-MS, were also reflected in the overall monosaccharide and N-glycan analyses. For example, more sialylated glycans were consistently observed on PRO1-468 compared to PRO1-471 by LC-MS and corresponded to the greater overall abundance of Neu5Ac in the HPAEC-PAD results (26.7 vs 21.2 mol Neu5Ac mol$^{-1}$ monomer, respectively).

Additionally, the shift to glycans with a greater number of antennae that was observed by LC-MS in PRO1-468 relative to PRO1-471 was manifested in the HPAEC-PAD results as a greater GlcN concentration in PRO1-468 (102.5 vs 95.6 mol GlcN mol$^{-1}$ monomer for PRO1-468 and PRO1-471, respectively). This same trend was also observed by HILIC-Fld analysis once the complexity of the N-glycan profiles was reduced by sialidase treatment (Supplementary Fig. 2).

Similar to HPAEC-PAD and HILIC-Fld, glycopeptide LC-MS also detected the trend towards lower relative intensities of di-galactosylated biantennary glycans relative to the corresponding non-galactosylated biantennary glycans for the *ST3GAL4* KO's (PRO1-469 and PRO1-470) when compared with the equivalent de-sialylated glycans from PRO1-468 and PRO1-471. The trend was less obvious due to the inherent complexity of the LC-MS results. Nevertheless, by averaging the proportion of Gal per site and summing for all sites (Supplementary Table 11, note that it is only an estimation and not a quantitative measure), it was apparent that the apparent galactosylation was lower for PRO1-469 and PRO1-470 (11.9% and 12.7%, respectively) than for PRO1-468 and PRO1-471 (15.0% and 14.0%, respectively). These

results are in remarkably good agreement with those obtained with HPAEC-PAD.

LC-MS analysis confirmed that fucosylated glycopeptides were absent from the digests of the spike proteins expressed in the *FUT8* KO cell lines (PRO1-468 and PRO1-470) as well as from the kifunensin-treated spike protein (PRO1-472). The complex-type glycans in the two remaining glycovariants (PRO1-469 and PRO1-471) were predominantly fucosylated. However, a noticeable proportion of the glycans on N1098 were afucosylated in both variants. The reason for this is not understood, although the same phenomenon has been observed by others[12]. It is notable that fucosylation levels were very similar for PRO1-469 and PRO1-471, implying that the presence or absence of sialylation does not impact fucosylation.

The same basic pattern of glycosylation was observed in all of the spike protein digests with the exception of PRO1-472. This suggests that knocking out *FUT8* (loss of core Fuc) and *ST3GAL4* (loss of 2–3-linked Neu5Ac) minimally disrupt glycan maturation and processing in the trans-golgi. As expected, all of the N-glycosylation sites on PRO1-472 (kifunensin-treated spike protein) were occupied with high-mannose glycans, with Man9 being the predominant glycan species.

**Glycosylation analysis of spike protein from independent production batches**. Three independent spike protein batches (PRO1-392, PRO1-394, and PRO1-412) produced from the same CHO pool were characterized using the neutral monosaccharide and sialic acid assays to determine if significant differences in monosaccharide content could be detected between them. The three batches were expressed in the same cell line (CHO[55E1]) and under the same conditions although harvesting was done on different days. As such, the glycan content and composition were expected to be relatively similar for all three batches, providing an opportunity to assess if the HPAEC-PAD monosaccharide assays could be sensitive enough to detect subtle variations in glycosylation. Here again, the monosaccharide results were compared with those from the HILIC-Fld and LC-MS assays.

**Monosaccharide analysis by HPAEC-PAD**. Consistent levels of Fuc and Man were observed between all batches. However, GlcN and Gal content was higher in PRO1-392 compared to PRO1-394 and PRO1-412 (Fig. 6a, Supplementary Table 6). Also, Neu5Ac content was significantly higher for PRO1-392 (16.5 mol Neu5Ac mol$^{-1}$ protein) compared to PRO1-394 (10.8 mol Neu5Ac mol$^{-1}$ protein) and PRO1-412 (8.7 mol Neu5Ac mol$^{-1}$ protein) (Fig. 6b,

Supplementary Table 6). We looked at the production record for each batch to better understand the variation in Neu5Ac content, (Supplementary Table 12). A range of factors can affect sialylation during fed-batch processes, including cell viability and/or fed-batch duration[40]. Indeed, cell viability at harvest time for PRO1-392 (96.5%) was slightly higher compared to that of PRO1-394 (95.2%) while PRO1-412 had the lowest viability (94.3%) (Supplementary Table 12). A possible explanation for the higher Neu5Ac content of PRO1-392 could be that the culture was harvested earlier than for the other two (day 7 for PRO1-392, day 10 for PRO1-412, and day 11 for PRO1-394). It is possible that more sialidase was released from dead cells and/or had more time to act on the sialylated glycans over the longer culture durations.

**HILIC-Fld analysis of released N-glycans**. The HILIC-Fld chromatograms of PRO1-394 and PRO1-412 are very similar, although Man5 is the dominant peak (14%) in PRO1-394 while FA2G2 is the dominant peak (14.5%) in PRO1-412 (Fig. 7). The chromatogram of PRO1-392 is distinct in having a Man5 peak of much smaller relative intensity (5.7%) than for the other two samples (14% for PRO1-394 and 11.8% for PRO1-412, respectively). Interestingly, the HPAEC-PAD monosaccharide assay did not detect a significant difference in Man content between PRO1-392 and the other batches. This was anticipated as Man is a component of all N-glycans and so a reduction in high-mannose glycan content is offset, at least partially, by an increase in other N-glycan species that contain 3 Man residues each. Finally, the dominant peak in PRO1-392 is the mono-sialylated FA2G2S1 glycan (16.6%), which suggests the highest level of sialylation than to the other two batches. Here again, this is consistent with the data from the HPAEC-PAD monosaccharide assays.

**LC-MS glycopeptide analysis**. Reliable LC-MS glycopeptide results were obtained for 20 of the 22 N-glycosylation sites for the three independent production batches (Supplementary Data 2, Supplementary Table 9). Here again, due to poor signal intensities and interference from other ions, glycopeptides could not be reliably identified and quantified for N709 or N1134 in all three batches.

Glycopeptide relative abundances revealed close similarity between PRO1-394 and -412. The PRO1-392 complex-type glycans were more sialylated and galactosylated than the other two batches and also possessed a larger proportion of tri- and tetra-antennary glycans (Supplementary Tables 9 and 10). This is

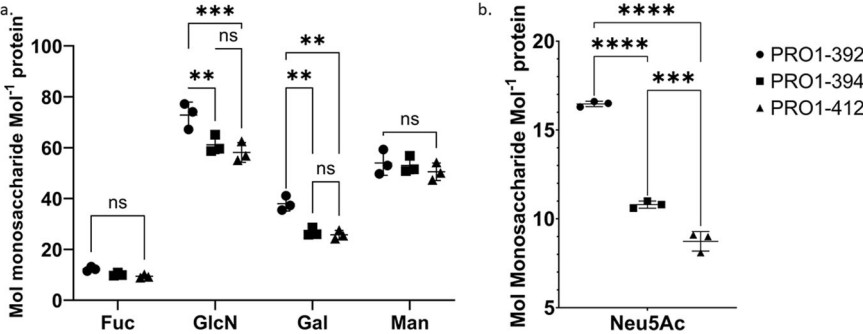

**Fig. 6 Monosaccharide analysis via HPAEC-PAD of three independent spike batches.** Results reported in mol monosaccharide mol$^{-1}$ monomer (MW = 143,230.6 Da). Results are the means of three triplicate injections and three triplicate reactions. Error bars are the standard deviation observed between the triplicate hydrolyses values. The data were analyzed using two-way ANOVA, Tukey's multiple comparisons test, comparing the content of each monosaccharide type between variants. Not all comparisons are shown. For full comparison see Supplementary Table 7. **a** Fuc, GlcN, Gal, and Man content obtained after TFA hydrolysis. **b** Neu5Ac content obtained after enzymatic hydrolysis. The data were analyzed using a one-way ANOVA, Tukey's multiple comparisons test, showing all comparisons. For exact *p* values see Supplementary Table 8. *P* values: <0.002 (**), <0.0002 (***) and <0.0001(****).

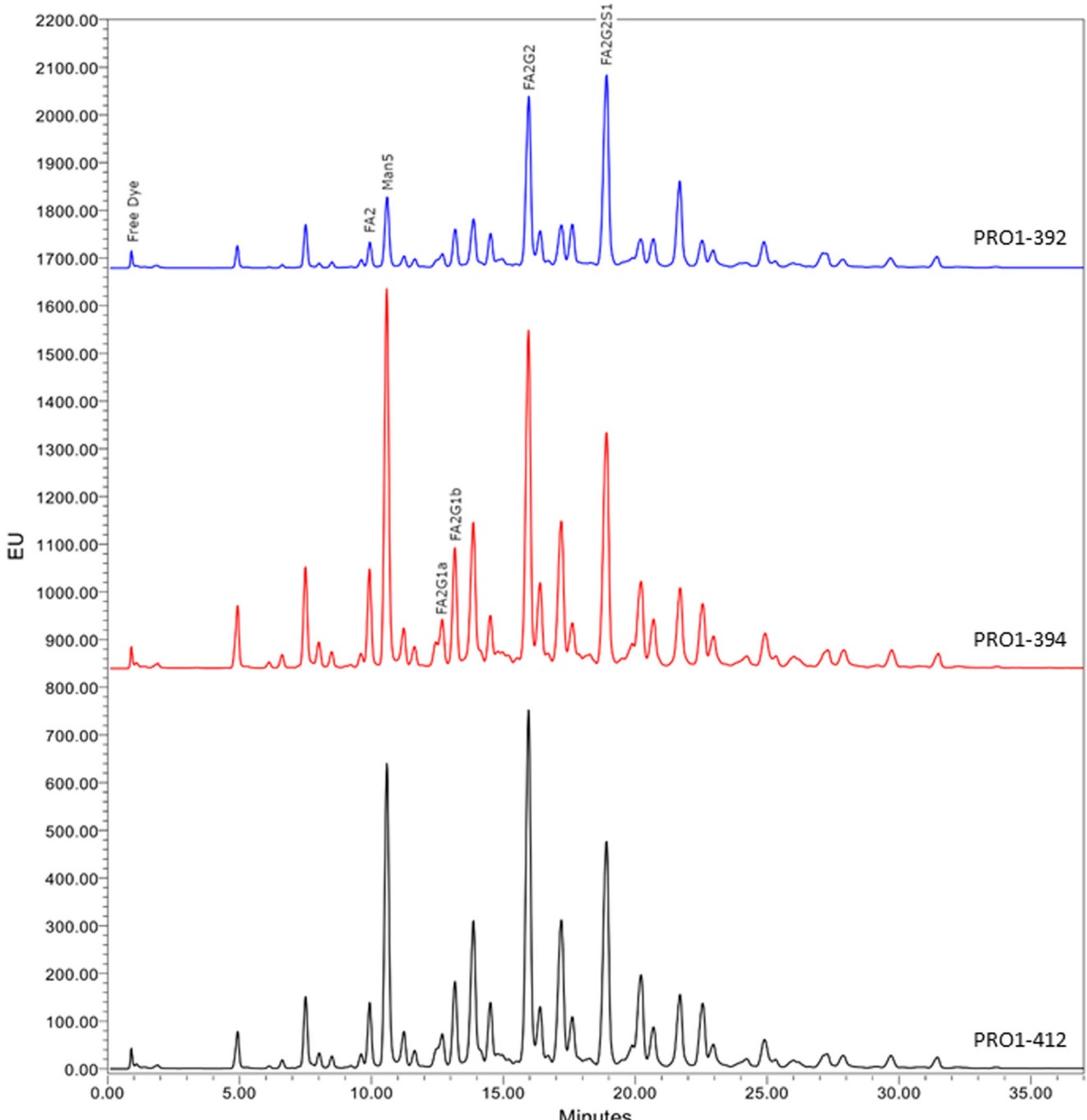

**Fig. 7 Comparison of HILIC-UPLC-Fld chromatograms of 3 batches of the spike protein produced in CHO pools.** The three HILIC traces for samples PRO1-392 (blue), PRO1-394 (red), and PRO1-412 (black) with offsets on the *y* axis to facilitate comparison and no offsets on the *x* axis are depicted.

consistent with data generated by both HPAEC-PAD and HILIC-Fld. In addition, the relative abundance of high-mannose glycopeptides was noticeably lower in PRO1-392 compared with the other two batches. As mentioned previously, the change in overall Man abundance in PRO1-392, as determined by monosaccharide analysis, is subtle due to the lower high-mannose glycan content being offset by the increase in complex and hybrid glycans, both of which contain mannose. This makes it difficult for the HPAEC-PAD monosaccharide assays to detect a significant change. On the other hand, increases in the Gal and GlcNAc content of PRO1-392 were detected by monosaccharide analysis and are indicative of the increased abundance of larger and/or more abundant complex-type glycans in PRO1-392.

## Discussion

For the reasons described previously, established assays struggle with the routine monitoring of the complicated and heterogenous glycans found on highly glycosylated antigens such as the SARS-CoV-2 spike glycoprotein. This can lead to gaps in our ability to compare glycosylation across multiple lots and in our understanding of how variation in glycosylation affects immunogenicity. Furthermore, there is little established guidance available for characterizing highly complex glycoproteins, and early discussions with regulatory authorities are usually required in order to determine the analytical package required[27].

The HPAEC-PAD monosaccharide assays are fast and sample preparation is simple, with no purification and derivatization steps

(Fig. 1). Furthermore, the results can be obtained in a few days without complex processing or calculations. Much effort has been put into addressing issues relating to the reliability of the HPAEC-PAD instrumentation[27]. In our experience, the technology has proven to be both reliable and robust. The goal of this investigation was to determine if batch-to-batch variations in spike protein glycosylation could be monitored effectively using the HPAEC-PAD monosaccharide assays. Two sets of engineered spike proteins, one with major and the other with minor differences in N-glycosylation, were analyzed by HPAEC-PAD and the results were compared with those obtained by glycan HILIC-Fld and glycopeptide LC-MS.

Glycan analysis by HILIC-Fld is regularly used for N-glycan batch monitoring of recombinant immunoglobulin Gs as their glycosylation profile is usually relatively simple. This simplicity stems from usually having just one N-glycosylation site on each heavy chain that is modified with biantennary glycans displaying limited variability. The HILIC-Fld assay requires the release, labeling, and purification of the glycan modifications, a process that is more elaborate than for HPAEC-PAD, Importantly, the HILIC-Fld data can be challenging to interpret for glycoproteins with multiple and heterogeneous N-glycans due to substantial peak overlap and the lack of suitable glycan standards for identification. As a result, it is often not possible to draw a precise picture of the glycosylation on these proteins. For example, many of the glycans released from the spike proteins used in this study could not be identified by HILIC-Fld and so it was not possible to determine their overall level of fucosylation or sialylation. In contrast, these were simple measurements to make via HPAEC-PAD.

The monosaccharide results generated via HPAEC-PAD were also compared with glycopeptide LC-MS data. Bottom-up LC-MS is typically used for the in-depth characterization of complex proteins and has the important advantage of being able to investigate glycosylation at individual sites. However, it is a slow, complex assay involving many processing and analysis steps (Fig. 1). Four different proteases were used to maximize the likelihood that each N-glycosylation site could be isolated on its own on a peptide that was compatible with LC-MS. As such it is unsuitable for the rapid and routine batch-to-batch monitoring of complex glycoproteins such as the SARS-CoV-2 spike. Furthermore, the resulting glycopeptide data sets are very complex which can often make it difficult to make important comparisons between clones or batches. For example, the lower galactose content of the non-sialylated spike variants was readily observed by the HPAEC-PAD monosaccharide analysis but was easy to overlook in the complex glycopeptide LC-MS data.

In conclusion, we demonstrate that it is possible to employ the HPAEC-PAD monosaccharide assays to monitor batch-to-batch changes in glycosylation for the SARS-CoV-2 spike glycoprotein antigens used for vaccine development. The results compared well with those obtained by glycan HILIC-Fld and glycopeptide LC-MS. In addition, we demonstrated that monosaccharide analysis could detect subtle glycosylation differences between batches of the same antigen expressed in the same cell line. Importantly, the monosaccharide assays were noticeably faster to perform and the results were quantitative, reproducible, and easy to interpret. As such, the monosaccharide assays have many of the features required for use in a QC environment. It can be very challenging to fully analyze glycan post-translational modifications and, certainly, HILIC-Fld and LC-MS are more suitable for the in-depth analysis of complex glycoproteins. Nevertheless, the HPAEC-PAD monosaccharide assays appear to be the better choice for monitoring batch-to-batch glycosylation of complex glycoproteins.

Ultimately, the strategic implementation of these monosaccharide assays could shorten the time it takes to develop a vaccine, an important consideration when responding to a future pandemic.

## Methods

### Production of the glycovariants (PRO1-468, PRO1-469, PRO1-470, PRO1-471, and PRO1-472).
Codon optimized DNA sequence encoding the ancestral (Wuhan), Genbank accession number: MN908947, SARS-CoV-2 spike protein[32], was cloned into the pTT5® vector[41] by Genscript, The sequence of the newly designed construct was verified by Sanger DNA sequencing.

Different KO CHO cell lines, including afucosylated (F15), asialylated (S9) as well as afucosylated and asialylated (dKO2) cells, all derived from CHO55E1 WT cells[32] using CRISPR/Cas9[42] (Supplementary Method 1), were used for protein expression. High-density transfection and post-transfection culture of cells were performed (Supplementary Method 2)[43]. Cells were cultured in a chemically-defined proprietary media formulation supplemented with 4 mM L-glutamine and incubated in shake flasks (Corning, NY, USA) under agitation (120 rpm) at 37 °C and with 5 % $CO_2$. Two days prior to transfection, cells were seeded at $1 \times 10^6$ cells $mL^{-1}$ in the same media to achieve a cell density of $\sim 8 \times 10^6$ $mL^{-1}$ on the day of transfection. Right before transfection, cells were diluted with 25% fresh media, and dimethylacetamide was added to 0.083 % (v $v^{-1}$). polyethylenimine -Max (Polysciences) was used to transfect cells at a DNA:polyethylenimine ratio of 1:7 (w:w) and plasmid DNA final concentration was 28 μg $mL^{-1}$ in cell culture media. The transfected DNA was a mix of 85% of the pTT5-spike construct, 10% pTT-Bcl-XL (anti-apoptotic effector), and 5% pTT-GFP were indicated, 10 μM (final) kifunensine (Toronto Research Chemicals, cat# K450000) was added to the culture 5 h post-transfection. At 24 h post-transfection, cells were moved to a 32 °C incubator and supplemented with Anti-Clumping Supplement (1:500 dilution) (Irvine Scientific) and Feed 4 (2.5% v $v^{-1}$) (Irvine Scientific). At 5 days post-transfection, cultures were fed with 5% of Feed 4, and additional Glc was added when needed to maintain a minimal concentration of 17 mM. Cell supernatants were collected at 7 days post-transfection.

Recombinant spike glycovariants were purified using AVIPure-COV2S resins (Avitide, Lebanon, New Hampshire) according to the manufacturer's instructions. Briefly, 0.2 μm-filtered supernatants were loaded on columns equilibrated with calcium- and magnesium-free Dulbecco's PBS (DPBS). After the initial wash with 10 column volumes of DPBS, proteins were eluted using a solution containing 50 mM Bis-Tris and 1 M Arginine-HCl at pH 6.0. The fractions containing eluted proteins were pooled and the elution buffer was buffer-exchanged for DPBS, using desalting columns (GE Healthcare). Purified proteins were visualized using SDS-Page (Supplementary Fig. 3) and quantified by spectrophotometry ($A_{280}$) using extinction coefficients calculated based on their amino acid composition.

### Production of spike from stable CHO pool PRO1-392, PRO1-394, and PRO1-412).
Batches of SARS-CoV-2 spike protein stably produced from a stable CHO pool (named PRO1-394, PRO1-392, and PRO1-412), were generated independently from a unique frozen cell vial (Supplementary Method 3)[44,45]. Culture conditions/characteristics for each batch are described in Supplementary Table 12.

**Analyses**. For analysis design, see Supplementary Method 4.

### HPAEC-PAD measurements
*Neutral sugar analysis.* The dried protein sample (25 μg) was heated, in triplicate with TFA (160 μL, 5 M) (Sigma, cat# 91707) in an oven at 100 °C for 2 h. The sample was allowed to cool and then was evaporated to dryness (Genevac EZ-2 Elite). The sample

was then re-suspended in water (125 μL), centrifuged at 14,000 rpm for 2 min, and then diluted with water 1:20 (v:v). 10 μl injections were eluted with a stepwise gradient of NaOH and were repeated in triplicate on Dionex ICS-6000 system equipped with a CarboPac PA20 column (3 × 150 mm, 060142), CarboPac PA20 guard column (3 × 30 mm, 060144), a BorateTrap column (4 × 50 mm, 047078) and a pulsed amperometric detector[23].

*Sialic acid analysis*. The dried protein sample (25 μg) was treated in triplicate with 5× Buffer B (10 μl) and Sialidase A (2 μL) (Agilent, GK80040) and adjusted to a final concentration of 0.48 μg/μL. The sample was kept at 37 °C for 18 h. The sample were then spun (14,000 rpm, 2 min), diluted 1:20 water, and 25 μL triplicate injections were eluted with a linear gradient of NaOH/ sodium acetate buffer on a Dionex ICS-6000 system equipped with a CarboPac PA20 column (3 × 150mm, 060142), CarboPac PA20 guard column (3 × 30 mm, 060144) and a pulsed ampero-metric detector[46].

Both neutral sugar and sialic acid data collection and analysis have been performed with Chromeleon 7 (Thermofisher Scientific). The data was processed using GraphPad Prism 9.0.2.

**HILIC-Fld**. Glycans were released from the glycoproteins using Rapid™ PNGase F (New England BioLabs, cat# P0710) at 50 °C for 10 min, after a 2 min at 80 °C denaturation step. The PNGase F and protein were removed using a Discovery Glycan SPE 50 mg column (Millipore-Sigma, cat# 55465-U), and the glycans were evaporated to dryness under vacuum. The glycans were then labeled with 2-AB (Sigma-Aldrich, cat# PP0520)[37]. The excess dye was removed on PD MiniTrap G-10 (Cytiva, cat# 28-9180-10). The sample was divided into two aliquots. One aliquot was used as is for glycan analysis while the other one was treated with a recombinant sialidase (MNV-01, expressed and purified in-house) prior to analysis, in order to identify peaks containing sialic acid. Glycans were analyzed by HILIC with fluorescent detection using an Acquity UPLC wide-pore glycoprotein Ethylene Bridged Hybrid amide, 300 Å, 1.7 μm, 2.1 × 150 mm column (Waters Corp., Cat# 176003702) heated to 60 °C and a flow rate of 0.5 mL min$^{-1}$. Glycans were eluted using a gradient of 100 mM ammonium formate, pH 4.5 (mobile phase A), and 100% acet-onitrile (mobile phase B), starting with an initial ratio of 22:78 and followed by a gradient of 44:56 over 43.5 min. The data was collected and analyzed using Xcalibur, version 4.4.16.14. Peaks were calibrated with a 2-AB labeled dextran ladder standard (Waters Corp., cat# 186006841) and compared to GU values in the database Glycobase. NIBRT (https://glycobase.nibrt.ie/ glycobase/show_nibrt.action). A recombinant broad-specificity sialidase was used to identify peaks containing sialic acid. The traces resulted in a non-quantifiable fingerprint.

**LC-MS/MS analyses**

*In-solution digestion*. Each of the variants (60 μg) was denatured with 0.1% Rapigest (Waters) in 150 μL of 50 mM ammonium bicarbonate (Sigma). Disulfide bonds were reduced with dithio-threitol (9.1 mM) (Sigma) at 80 °C for 10 min then alkylated with iodoacetamide (34.1 mM) (Sigma) at room temperature for 30 min. Excess iodoacetamide was quenched with 15 μL of 100 mM DTT (15 min at room temperature) to prevent over-alkylation. Samples were buffer-exchanged into 50 mM ammonium bicarbonate using 7 K Zeba spin-columns (Thermo Scien-tific), following product instructions, and then diluted to a concentration of 0.1 μg μL$^{-1}$ with ammonium bicarbonate (50 mM). Half of each sample (30 μg protein) was treated with PNGase F (3 U) (Sigma) and incubated at 37 °C overnight to deglycosylate. Both deglycosylated and untreated halves were

separately divided into four equal parts (7.5 μg each) that were treated with either trypsin (0.5 μg) (Promega), Glu-C (0.5 μg) (Promega), α-lytic protease (0.4 μg) (Sigma) or chymotrypsin (0.25 μg) (Sigma). The chymotrypsin digests were incubated overnight at 25 °C while the other digests were incubated over-night at 37 °C. All digests were acidified with a 1/20 volume of 10% TFA (Sigma), incubated at 37 °C for 40 min, and centrifuged at 13,000 rpm for 15 min to cleave and remove Rapigest. Each digest (2 μL) was diluted with 0.1% formic acid (10 μL) for injection. The trace

*LC-MS*. Peptide digests (10 μL, ~1.3 pmol) were injected on a C18 PepMap100 5 μm trap (Thermo Scientific) installed before a nanoACQUITY Ethylene Bridged Hybrid 1.7 μm column (100 μm × 100 mm C18, Waters) in an UltiMate 3000 nano-LC system (Thermo Scientific) and detected on an Orbitrap Eclipse Tribrid mass spectrometer equipped with an electrospray ioni-zation source (Thermo Scientific). The peptides were eluted with a gradient of 0.1% formic acid in ddH$_2$O (Mobile phase A) and 0.1% formic acid in acetonitrile (Mobile phase B). Peptides were eluted over a linear gradient of 0.2% to 40% mobile phase B over 89 min, 40% to 95% mobile phase B over 4 min, followed by re-equilibration, with a flow rate of 0.5 μL min$^{-1}$. MS spectra were acquired on the Orbitrap from 350 to 2000 *m/z* in positive electrospray ionization mode at a 120 K resolution. The most intense ions (threshold = 2e6, dynamic exclusion = 25 sec) were selected in the quadrupole for HCD-MS/MS (isolation win-dow = 1.6 *m/z*) and a fixed HCD activation energy was selected based on peptide *m/z* and charge state using a decision tree. MS/ MS spectra were acquired in the Orbitrap at 15 K resolution in profile mode with a 1 sec cycle time. The data was recorded using Xcalibur, version 4.4.16.14.

*Data analysis*. The data files were manually inspected to identify (glyco)peptides covering each putative N-glycosylation site (Supplementary Table 4). Peptides covering 2 or more N-glycosylation sites were rejected to ensure each site could be isolated on its own. Glycopeptide identity was confirmed by manual inspection of the HCD-MS/MS spectrum and expected retention time windows for related glycopeptides were defined from the manually identified glycopeptides. The data were pro-cessed using GlycoPIQ v2.1, an in-house software (see Supple-mentary Methods 5) that allows the identification and quantification of a large number glycopeptides within nano-LC-MS/MS data sets. The data was then visualized using Excel.

**Reporting summary**. Further information on research design is available in the Nature Portfolio Reporting Summary linked to this article.

## Data availability

The data are included in the paper and/ or its supplementary information files. The authors declare that the data supporting the findings of this study are available within the paper and its supplementary information files (Supplementary Data 1 and Data 2).

## Code availability

In-house software GlycoPIQ v2.1 is available upon request and is similar to other free data processing software such as FragPipe. https://fragpipe.nesvilab.org/docs/tutorial_ glyco.html.

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

## Acknowledgements

We would like to thank Sam Williamson and Luc Tessier for the preparation and maintenance of the LC-MS instrumentation. A special thanks to Simon Lord-Dufour for the generation of the stable CHO pool and fed-batch productions. The expert contribution of Christian Gervais, Louis Bisson, Brian Cass, and Guillaume Arthus-Cartier for spike purification by IMAC is acknowledged. We would like to acknowledge Dr. Andrew Cox for his constant support and fruitful discussions during this project. We also acknowledge the NRC's Pandemic Response Challenge Program as a funding source for this work.

## Author contributions

J.S., M.G., J.K., and Y.D. secured funding for the work. J.S., M.G., and J.K. wrote the manuscript. J.S. and I.S. supported the development of the Neu5Ac assay. J.S. and F.S.M. developed the neutral sugar assay. F.S.M. performed the analyses on the HPAEC-PAD and J.S. performed the data analyses of neutral sugar and Neu5Ac analyses. I.K. and S.P. produced and purified the glycoproteins under Y.D.'s supervision. M.G., D.B., and M.F.G. performed the HILIC-Fld analyses. J.K., A.S., A.H., and A.R. performed the LC-MS analyses.

## Competing interests

The authors declare no competing interests.
