## [Peer Review File · Communications Chemistry]

Reviewers' comments:

Reviewer #1 (Remarks to the Author):

The manuscript was generally well written and easy to read. The authors presented a monosaccharide analysis method that can be quickly applied to glycan characterization and batch-to-batch consistency evaluation in the development and production process of biopharmaceuticals such as vaccines. Comparison of analytical data of monosaccharides, glycans, and glycopeptides obtained from SARS-CoV-2 spike proteins produced in various KO cell lines to confirm the usefulness of this assay is scientifically meaningful and interesting.

However, the HPAED-PAD monosaccharide analysis, proposed as an assay that can sensitively detect differences in glycosylation between batches in vaccine development, has traditionally been used to analyze the quality characteristics and consistency of various biopharmaceuticals, including antibodies. Therefore, the method lacks novelty and its limitations are clear. These aspects should be fully considered and described in the manuscript.

More specifically

1) SARS-CoV-2 spike proteins have been reported to have glycan modifications such as sulfation in addition to typical monosaccharides (refs 1, 2). Although the monosaccharide analysis method proposed by the author has limitations in obtaining information on glycan modification, this aspect is not sufficiently discussed in the manuscript. In fact, it is known that glycan modifications such as O-acetylation (OAc), phosphorylation, and sulfation in glycosylation are related to improved efficacy and function of biopharmaceuticals. (refs 3 ,4).

Ref 1. Anal. Chem. 2020, 92, 21, 14730–14739, Comprehensive Analysis of the Glycan Complement of SARS-CoV-2 Spike Proteins Using Signature Ions-Triggered Electron-Transfer/Higher-Energy Collisional Dissociation (ETHcD) Mass Spectrometry

Ref 2. Cell Host Microbe 2020, 28 (4), 586-, Virus-Receptor Interactions of Glycosylated SARS-CoV-2 Spike and Human ACE2 Receptor

Ref 3. Drug Discov Today 2007, 12 (7-8), 319-326, Sialic acids: carbohydrate moieties that influence the biological and physical properties of biopharmaceutical proteins and living cells

Ref 4. Anal Chem 2019, 91 (9), 6064-6071, Comprehensive Characterization of Biotherapeutics by Selective Capturing of Highly Acidic Glycans Using Stepwise PGC-SPE and LC/MS/MS

2) In addition, glycan has various isomers, and specific isomers are directly involved in the safety and efficacy of biotherapeutics. For example, the glycan corresponding to the composition of Hex5HexNc4Fuc1 may be a G2F glycan expressed in an antibody or may be an alpha-gal structure glycan that affects immunogenicity. However, monosaccharide analysis alone cannot identify the glycans of the two structures mentioned above. That is, despite the advantage of absolute quantification of monosaccharide units in batch-to-batch equivalent comparisons of glycosylation, the inherent limitations of monosaccharide analysis should be addressed in the Results and Discussion section.

3) The number of identified glycans is too small compared to the peaks detected in the glycan analysis, as shown in Figure 3 and Figure 7. As mentioned, glycan identification can be challenging, but results of

glycan profiling of SARS-COV-2 proteins have already been reported. In particular, comparing monosaccharide results for only the few key glycans identified and saying that there is a high correlation between glycan profiling data and monosaccharide quantification is considered a jump in results. Quantitative data of glycan expression, total glycan profiling results, and quantitative values of monosaccharides should be considered integrally.

4) It is mentioned that the results of glycopeptide analysis and monosaccharide analysis are correlated, and glycopeptide analysis is difficult and requires a lot of time and effort, making it unsuitable for batch-to-batch glycosylation evaluation through fast and routine analysis. Glycopeptide assay is an approach to evaluate the macro- and micro-heterogeneity of glycosylation by each site, and it is true that analysis and data interpretation are more difficult than other methods. Although the time and effort required for sample processing, analysis, and data interpretation are different depending on the information obtained from each method, the authors seem to have exploited the limitations of each method to emphasize the suitability of the monosaccharide method. Therefore, it seems that the result and discussion part need to be appropriately modified.

Reviewer #2 (Remarks to the Author):

Major Comments:

1. The subject is interesting, and the general approach seems acceptable though not novel. However, lack of experimental details makes it difficult for other researchers to re-produce and/or follow up.
 - 1) Page 21, Neutral sugar analysis: please specify the sample concentration of the 25 ug protein and clarify the meaning of the 5M TFA. Is this "5M" the final TFA concentration or you used 160 uL of 5M TFA?
 - 2) Page 22, Sialic Acid Analysis: Please specify the concentration of the 25 ug of protein sample.
 - 3) Page 22, HILIC-Fld: please clarify the use of sialidase for HILIC-Fld of release glycans. It seems no sialidase used in the glycan release procedure.
 - 4) Page 23: In-solution digestion: Please specify the sample concentration and volume ratio sample:RapiGest solution. Is the 0.1% RapiGest the concentration before or after mixing with protein samples? Please also specify how much DTT was added for quenching. In addition, please justify why it is necessary use for four enzymes in the discussion.
2. You have shown that the monosaccharide assays were significantly faster to perform, and the results were quantitative, reproducible and easy to interpret. I would suggest adding comments on its weaknesses and gap between the monosaccharide assays and other HILIC-Fld and LCMS.
3. You used HILIC-Fld for release glycan analysis. In recent years, HILIC-Fld coupled with MS is commonly used for glycan characterization. You have the LC-MS capability, I am wondering why you did not use HILIC-Fld/MS for released glycan analysis. Otherwise, you can assign more peaks in Fig 3 and Fig. 4 so that it can do a better comparison between assays.
4. Page 12, LC-MS glycopeptide analysis: Please explain why glycopeptides covering N134 was not detected. Please confirm if you only detected the unglycosylated peptide covering this site and did not see any difference all digests of intact and deglycosylated samples using all four enzymes.
5. Page 18, HILIC-Fld: Please explain more about the discrepancy of Man 5 results between HILIC-Fld vs

HPAEC-PAD and LC-MS vs. HPAEC-PAD. Why was the discrepancy anticipated?

Minor Comments:

1. Page 8, Figure Caption: The legend symbols for PRO 1-470 and PRO 1-471 in Panel b do not match the Figure. Redrawn the symbols is needed.
2. Page 10, Figure 10: Please clarify if all overlaid chromatograms are at normalized scale and the offset for the overlays.
3. Page 11, Figure 4: same as above.
4. Page 17, Figure 5: same as above.
5. Page 20, Conclusion: You may emphasize that HPAEC-PAD based these monosaccharide assays is QC friendly while LC-MS and HILIC-Flu/MS assays are good for in-deep characterizations.

27nd of June 2023

Re: Response to reviewers, Manuscript COMMSCHEM-22-0551B

Dear reviewers

We appreciate the time and effort you have dedicated to reviewing our manuscript titled “Simplifying glycan monitoring of complex antigens such as the SARS-CoV-2 spike to accelerate vaccine development”. Your insightful feedback has been invaluable in improving the quality and clarity of our work.

Reviewer 1:

Thank you for your thorough evaluation of our manuscript. Your comments on the novelty of the HPAEC-PAD and its fair differentiation from other methods, such as HILIC-Fld and LC-MS, have been further elaborated in our manuscript. HPAEC-PAD is not novel, but its application to complex protein batch-to-batch characterization is. In light of this, we have edited our manuscript to reflect these considerations.

Reviewer 2:

We appreciate your thorough evaluation of our manuscript and the need for clarifying our experimental methods. We have corrected experimental oversights in our manuscript.

We have included below a point-by-point response to each comment raised. We appreciate your thoughtful review of our work, and are convinced that the revised version contributes to a highly impactful study, with interest to the Communication Chemistry readership.

Yours sincerely

Janelle Sauvageau (PhD) She/Her/Elle
Research Officer,
National Research Council of Canada
100 Sussex Dr.
Ottawa
K1N 5A2
Canada
001-613-408-6241

Response to reviewer 1 (Answers italicized and bold, with line numbers corresponding to the annotated version):

The manuscript was generally well written and easy to read. The authors presented a monosaccharide analysis method that can be quickly applied to glycan characterization and batch-to-batch consistency evaluation in the development and production process of biopharmaceuticals such as vaccines. Comparison of analytical data of monosaccharides, glycans, and glycopeptides obtained from SARS-CoV-2 spike proteins produced in various KO cell lines to confirm the usefulness of this assay is scientifically meaningful and interesting.

Thank you for your kind comment

1. However, the HPAEC-PAD monosaccharide analysis, proposed as an assay that can sensitively detect differences in glycosylation between batches in vaccine development, has traditionally been used to analyze the quality characteristics and consistency of various biopharmaceuticals, including antibodies. Therefore, the method lacks novelty and its limitations are clear. These aspects should be fully considered and described in the manuscript.

HPAEC-PAD is not novel, however, the application of this method to the characterization of complex glycoproteins is novel. There is little in the literature in relation to recommendations regarding the analysis of complex protein antigens. Indeed, most publications relate to the analysis of mAbs which have significantly simpler glycoform profiles. One of our aims is to address this gap in the literature by starting a conversation about alternate ways to characterize biotherapeutic proteins with complex glycosylation. Currently this discussion occurs outside of the scientific literature between regulators and companies and the wider community gains little in terms of new knowledge and lessons learned. (Butré, C. I. & Delobel, A. in Carbohydrate Analysis by Modern Liquid Phase Separation Techniques (Second Edition) (ed Ziad El Rassi) 761-814 (Elsevier, 2021).):

It is interesting to note that HPAEC-PAD was used in the past to analyze glycoproteins. It waned in popularity due to the development of techniques such as HILIC-FLD which are better suited for the analysis of the simpler glycan populations found on mAbs. However, HPAEC-PAD has benefited from many technology improvements in recent years and we believe that it can make an important contribution to the characterization of complex glycoproteins, especially in a scenario where a vaccine is needed urgently and development time needs to be as short as possible.

We are aware of the limitations of HPAEC-PAD and of monosaccharide analysis as a means of thoroughly characterizing the glycosylation of biotherapeutic proteins. We have ensured that this is more fully discussed in the manuscript. However, please note that HPAEC-PAD is not meant to be a substitute for detailed glycosylation analysis using established assays. Instead, we propose that it can be used to monitor for batch-to-batch changes in glycosylation that could signal a significant difference in the quality of the complex glycoproteins being produced. This scenario would trigger a more rigorous analysis using established glycosylation assays.

We have thus added this comment in the introduction, p.4, line 92:

“HPAEC-PAD was used in the past to analyze glycoproteins. It waned in popularity due to the development of techniques such as HILIC-FLD which are better suited for the analysis of the simpler glycan populations found on mAbs. In addition, the robustness of the method relative to other assays was questioned²⁶⁻²⁸. However, recent technological advances such as the introduction of disposable electrodes to eliminate fouling as well as the use of calibration curves to normalize the response differences between monosaccharides, have addressed most of these criticisms.”

We have also added the following sentence to the Discussion, p.20 line 437:

“Much effort has been put into addressing issues relating to the reliability of the HPAEC-PAD instrumentation. In our experience, the technology has proven to be both reliable and robust.”

2. More specifically

1) SARS-CoV-2 spike proteins have been reported to have glycan modifications such as sulfation in addition to typical monosaccharides (refs 1, 2). Although the monosaccharide analysis method proposed by the author has limitations in obtaining information on glycan modification, this aspect is not sufficiently discussed in the manuscript. In fact, it is known that glycan modifications such as O-acetylation (OAc), phosphorylation, and sulfation in glycosylation are related to improved efficacy and function of biopharmaceuticals. (refs 3 ,4).

Ref 1. Anal. Chem. 2020, 92, 21, 14730–14739, Comprehensive Analysis of the Glycan Complement of SARS-CoV-2 Spike Proteins Using Signature Ions-Triggered Electron-Transfer/Higher-Energy Collisional Dissociation (ETHcD) Mass Spectrometry

Ref 2. Cell Host Microbe 2020, 28 (4), 586-, Virus-Receptor Interactions of Glycosylated SARS-CoV-2 Spike and Human ACE2 Receptor

Ref 3. Drug Discov Today 2007, 12 (7-8), 319-326, Sialic acids: carbohydrate moieties that influence the biological and physical properties of biopharmaceutical proteins and living cells

Ref 4. Anal Chem 2019, 91 (9), 6064-6071, Comprehensive Characterization of Biotherapeutics by Selective Capturing of Highly Acidic Glycans Using Stepwise PGC-SPE and LC/MS/MS

Thank you for this observation and for these interesting references. We wish to point out that the sulfation levels observed in references 1 and 2 were very subtle and would be hard to detect using any of the established assays. Somewhat related to this, it has been shown that tyrosine sulfation of recombinant proteins is highly dependent on the cell line used and can be attenuated using modifiers such as sodium chlorate (Liu, R. et al., Modulating tyrosine sulfation of recombinant antibodies in CHO cell culture by host selection and sodium chlorate supplementation. Biotechnology Journal 16, 2100142 (2021). [https://doi.org:https://doi.org/10.1002/biot.202100142](https://doi.org/https://doi.org/10.1002/biot.202100142)). More specifically, while it is not certain yet that HPAEC-PAD can detect sulfated sugars, it has been shown that HPAEC-PAD can detect phosphorylated mannose residues (Zhou, Q., Kyazike, J., Edmunds, T. & Higgins, E. Mannose 6-phosphate quantitation in glycoproteins using high-pH anion-exchange chromatography with pulsed amperometric detection. Anal Biochem 306, 163-170 (2002). <https://doi.org:10.1006/abio.2002.5703>).

To clarify these points, we added the following comment to the introduction p.4 line 98:

“Moreover, it has been shown that HPAEC-PAD can detect monosaccharides with unusual modifications such as mannose-6-phosphate which has been found on some glycoproteins. It has to be determined yet if HPAEC-PAD can identify sugars with other modifications such as sulfation, a modification observed in low levels in the literature for the spike protein in HEK-293 cells^{13,29}. “ Please note that we did not add a comment related to unnatural modifications as discussed in reference 3 and 4 as it is not directly relevant to this particular work. It may be possible to detect unusual monosaccharides by HPAEC-PAD but this has not been investigated as of yet.

3. In addition, glycan has various isomers, and specific isomers are directly involved in the safety and efficacy of biotherapeutics. For example, the glycan corresponding to the composition of Hex5HexNc4Fuc1 may be a G2F glycan expressed in an antibody or may be an alpha-gal structure glycan that affects immunogenicity. However, monosaccharide analysis alone cannot identify the glycans of the two structures mentioned above. That is, despite the advantage of absolute quantification of monosaccharide units in batch-to-batch equivalent comparisons of glycosylation, the inherent limitations of monosaccharide analysis should be addressed in the Results and Discussion section.

We agree with this criticism. Monosaccharide analysis by HPAEC-PAD is unable to provide the detailed information about glycan structure and diversity of the other assays investigated here. However, our argument is that the monosaccharide assays can quickly monitor for potential changes in the glycan profiles of complex glycoproteins which, in turn, would trigger additional and more in-depth analysis. We have added the following statement to the Discussion, p.21, line 480.

“It can be very challenging to fully analyze glycan posttranslational modifications and, certainly, HILIC-Fld and LC-MS are more suitable for the in-depth analysis of complex glycoproteins. Nevertheless, the HPAEC-PAD monosaccharide assays appear to be the better choice for monitoring batch-to-batch glycosylation of complex glycoproteins.”

4. The number of identified glycans is too small compared to the peaks detected in the glycan analysis, as shown in Figure 3 and Figure 7. As mentioned, glycan identification can be challenging, but results of glycan profiling of SARS-COV-2 proteins have already been reported. In particular, comparing monosaccharide results for only the few key glycans identified and saying that there is a high correlation between glycan profiling data and monosaccharide quantification is considered a jump in results. Quantitative data of glycan expression, total glycan profiling results, and quantitative values of monosaccharides should be considered integrally.

Thank you for this comment. Yes, relatively few glycan peaks in the HILIC-Fld traces have been labeled. We identified only those in which we have a high degree of confidence. Comparing with published glycan results for SARS-CoV2 glycoproteins has proven difficult as many of the peaks are

only partially resolved from one another. Nevertheless, it was possible to follow general trends in the glycan HILIC-Fld traces which could be correlated with the monosaccharide results. We have reviewed the manuscript in light of this comment and ensured that any claims made in relation to correlation of the monosaccharide results with those obtained for the glycan HILIC-Fld and glycopeptide LC-MS assays are as conservative as possible.

5. It is mentioned that the results of glycopeptide analysis and monosaccharide analysis are correlated, and glycopeptide analysis is difficult and requires a lot of time and effort, making it unsuitable for batch-to-batch glycosylation evaluation through fast and routine analysis. Glycopeptide assay is an approach to evaluate the macro- and micro-heterogeneity of glycosylation by each site, and it is true that analysis and data interpretation are more difficult than other methods. Although the time and effort required for sample processing, analysis, and data interpretation are different depending on the information obtained from each method, the authors seem to have exploited the limitations of each method to emphasize the suitability of the monosaccharide method. Therefore, it seems that the result and discussion part need to be appropriately modified.

Thank you for this insightful comment. The purpose of this manuscript is to highlight the potential of the HPAEC-PAD monosaccharide assays for routine monitoring of complex glycoprotein production batches. It is not our intention to purposely exploit potential weaknesses associated with the other techniques so as to promote the monosaccharide assays. Instead, the arguments outlined in this manuscript are based on experience acquired during the Covid-19 pandemic. Our established glycosylation assays really struggled to keep up with the demands of our vaccine research and production programs. The monosaccharide assays addressed the need for a rapid and reliable “first pass” assessment of the glycosylation of complex glycoprotein antigens. Nevertheless, the manuscript has been reviewed with this comment in mind and modified accordingly.

Reviewer #2 (Remarks to the Author):

Major Comments:

1. The subject is interesting, and the general approach seems acceptable though not novel.

The HPAEC-PAD is not a novel method, however, its application to the analysis of glycans from complex glycoproteins is novel. As noted recently by Butré and Delobel, discussions on complex glycoprotein glycan analyses occur however, not in the literature and why we wanted to close that gap: in Carbohydrate Analysis by Modern Liquid Phase Separation Techniques (Second Edition) (ed Ziad El Rassi) 761-814 (Elsevier, 2021). We have modified the introduction of the manuscript, p. 4, line 92 to reflect.

2. However, lack of experimental details makes it difficult for other researchers to re-produce and/or follow up.

Thank you for highlighting the following oversights. We have modified the manuscript accordingly.

1) Page 21, Neutral sugar analysis: please specify the sample concentration of the 25 ug protein and clarify the meaning of the 5M TFA. Is this "5M" the final TFA concentration or you used 160 uL of 5M TFA?

In this case, we used 25µg of dried protein. 5M is the final TFA concentration. We used 160 µL of 5M TFA. We have clarified this in the text, p. 23, line 527

2) Page 22, Sialic Acid Analysis: Please specify the concentration of the 25 ug of protein sample.

Here again, we used 25 µg of dried protein sample. The final concentration after adding buffer and sialidase A was 0.48 µg/µL. This has been clarified in the text, p. 23, line 539

3) Page 22, HILIC-FLD: please clarify the use of sialidase for HILIC-FLD of release glycans. It seems no sialidase used in the glycan release procedure.

The glycans were released from the spike glycoprotein using Rapid PNGase F. No sialidase was used in this release procedure. However, in order to aid in the interpretation of the HILIC-FLD chromatograms, a portion of the 2-AB derivatized glycans was desialylated prior to analysis using a sialidase that was produced in-house.

P.24, line 550 we edited to clarify: "The sample was divided into two aliquots. One aliquot was used as is for glycan analysis while the other one was treated with a recombinant sialidase (MNV-01, expressed and purified in-house) prior to analysis, in order to identify peaks containing sialic acid."

4) Page 23: In-solution digestion: Please specify the sample concentration and volume ratio sample:RapiGest solution. Is the 0.1% RapiGest the concentration before or after mixing with protein samples? Please also specify how much DTT was added for quenching. In addition, please justify why it is necessary use for four enzymes in the discussion.

The additional information regarding the sample concentration, Rapigest concentration and quenching DTT concentration has been added. The following justification for why 4 proteases are required has been added to the result section. P. 21, lane 461.

"Four different proteases were used to maximize the likelihood that each N-glycosylation site could be isolated on its own on a peptide that was compatible with LC-MS."

5) You have shown that the monosaccharide assays were significantly faster to perform, and the results were quantitative, reproducible and easy to interpret. I would suggest adding comments on its weaknesses and gap between the monosaccharide assays and other HILIC-FLD and LCMS.

We have added the following comment to the Introduction, p.4, line 92:

"HPAEC-PAD was used in the past to analyze glycoproteins. It waned in popularity due to the development of techniques such as HILIC-FLD which are better suited for the analysis of the simpler

glycan populations found on mAbs. In addition, the robustness of the method relative to other assays was questioned²⁶⁻²⁸. However, recent technological advances such as the introduction of disposable electrodes to eliminate fouling as well as the use of calibration curves to normalize the response differences between monosaccharides, have addressed most of these criticisms.”

We have also added the following sentences to the Discussion, p. 20, line 436:

“Much effort has been put into addressing issues relating to the reliability of the HPAEC-PAD instrumentation. In our experience, the technology has proven to be both reliable and robust.”

“It can be very challenging to fully analyze glycan posttranslational modifications and, certainly, HILIC-Fld and LC-MS are more suitable for the in-depth analysis of complex glycoproteins. Nevertheless, the HPAEC-PAD monosaccharide assays appear to be the better choice for monitoring batch to batch glycosylation of complex glycoproteins.”

6) You used HILIC-Fld for release glycan analysis. In recent years, HILIC-Fld coupled with MS is commonly used for glycan characterization. You have the LC-MS capability, I am wondering why you did not use HILIC-Fld/MS for released glycan analysis. Otherwise, you can assign more peaks in Fig 3 and Fig. 4 so that it can do a better comparison between assays.

This is a good point. We have dedicated UPLC-HILIC-Fld and LC-MS capabilities but, unfortunately, they are located at different geographical sites. Currently, we do not have the ability to directly interface UPLC-HILIC-Fld with MS. However, it is worth noting that UPLC-HILIC-Fld is still the “gold standard” assay that most biomanufacturing/bioanalytical facilities use for glycosylation analysis. As such, they are likely to encounter the same identification challenges when attempting to analyze complex glycoproteins.

7) Page 12, LC-MS glycopeptide analysis: Please explain why glycopeptides covering N134 was not detected. Please confirm if you only detected the unglycosylated peptide covering this site and did not see any difference all digests of intact and deglycosylated samples using all four enzymes.

A few glycopeptides for N1134 were detected in the alpha-lytic digests but they were relatively weak and not consistent from one sample to another. In addition, the GlycoPIQ analysis of the N1134 glycopeptides returned a high number of false positives, significantly more so than for the other N-glycosylation sites. For these reasons, we considered that the relative abundance results for this site to be unreliable. Interestingly, we encountered similar issues for the N709 glycopeptides (also alpha-lytic) but only for the second cohort of spike protein samples. Statements to this effect have been added to the manuscript, p. 12, lane 266 and p. 19, lane 402

8. Page 18, HILIC-Fld: Please explain more about the discrepancy of Man 5 results between HILIC-Fld vs HPAEC-PAD and LC-MS vs. HPAEC-PAD. Why was the discrepancy anticipated?

Thank you for this question. We have modified this section as follows, p.19, line 415.

“As mentioned previously, the change in overall Man abundance in PRO1-392, as determined by monosaccharide analysis, is subtle due to the lower high-mannose glycan content being offset by the increase in complex and hybrid glycans, both of which contain mannose. This makes it difficult for the HPAEC-PAD monosaccharide assays to detect a significant change. On the other hand, increases in the Gal and GlcNAc content of PRO1-392 were detected by monosaccharide analysis and are indicative of the increased abundance of larger and/or more abundant complex-type glycans in PRO1-392.”

3. Minor Comments:

1. Page 8, Figure Caption: The legend symbols for PRO 1-470 and PRO 1-471 in Panel b do not match the Figure. Redrawn the symbols is needed.

This has been redrawn

2. Page 10, Figure 10: Please clarify if all overlaid chromatograms are at normalized scale and the offset for the overlays.

3. Page 11, Figure 4: same as above.

4. Page 17, Figure 5: same as above.

We assume that the reviewer means Figure 3 (not 10), Figure 4 and Figure 5. In fact, the overlaid chromatograms did not need any scale normalization or retention time offset and thus no additional comments were added to that effect in the manuscript.

5. Page 20, Conclusion: You may emphasize that HPAEC-PAD based these monosaccharide assays is QC friendly while LC-MS and HILIC-Fld/MS assays are good for in-deep characterizations.

The following statement was added to the Discussion section, p.21, line 478:

“...Importantly, the monosaccharide assays were significantly faster to perform and the results were quantitative, reproducible and easy to interpret. As such, the monosaccharide assays have many of the features required for use in a QC environment. It can be very challenging to fully analyze glycan posttranslational modifications and, certainly, HILIC-Fld and LC-MS are more suitable for the in-depth analysis of complex glycoproteins. Nevertheless, the HPAEC-PAD monosaccharide assays appear to be the better choice for monitoring batch to batch glycosylation of complex glycoproteins.”

REVIEWERS' COMMENTS:

Reviewer #1 (Remarks to the Author):

The authors have revised and supplemented the manuscript with appropriate responses to comments to improve overall quality. Although the proposed method is not completely new, the topic of the paper is interesting and the authors clearly state the limitations of the method in the discussion section. Therefore, the manuscript is considered acceptable and I have no further comments.

Reviewer #2 (Remarks to the Author):

Thanks for making efforts to revise the manuscript, most of many comments have been addressed. I have a few more comments and hope that you can address them before acceptance for publication:

1. One of my original comments is to compare the monosaccharide assay with other glycan assays (HILIC-Fld and LC-MS) and comment on its weaknesses when emphasizing its advantages. You added some comments on p4 (line 92) and p20 (line 436). I would say that structure information is missing with monosaccharide assay, while HILIC-Fld provides glycan structures, LC-MS can provide locations of glycosylation. Please consider including this comment in your final manuscripts.
2. On P23, Line525: "(Agilent, GK80040) at 37°C for 18 h." This is not a complete sentence. It seems that you missed some information for sample preparation for Sialic Acid Analysis.
3. For Figures 3,4 and 7. You have Y-offsets for overlaid chromatograms. I mean that, using Fig.4 as an example, you have 350 y-offset, the y starts from 0 for the bottom chromatogram, but start from 350 for the overlaid chromatogram. You should add a note in the figure caption.

2nd of August 2023

Re: Response to reviewers, Manuscript COMMSCHEM-22-0551B

Dear reviewers,

We want to express our appreciation for reviewing our manuscript titled “Simplifying glycan monitoring of complex antigens such as the SARS-CoV-2 spike to accelerate vaccine development”. Your feedback and comments have been instrumental in improving the overall quality of our work.

Reviewer 1:

Thank you for your kind comments. Your acceptance of our manuscript is genuinely encouraging.

Reviewer 2:

Thank you for your time and effort in providing such a comprehensive review.

We have edited our manuscript to answer to the three remaining comments raised by reviewer 2. We truly appreciate your expertise and commitment to the peer review process, which has undoubtedly strengthened our research,

Yours sincerely

Janelle Sauvageau (PhD) She/Her/Elle
Research Officer,
National Research Council of Canada
100 Sussex Dr.
Ottawa
K1N 5A2
Canada
001-613-408-6241

Response to reviewer 2 (Answers italicized and bold, with line numbers corresponding to the final version):

1. One of my original comments is to compare the monosaccharide assay with other glycan assays (HILIC-FLD and LC-MS) and comment on its weaknesses when emphasizing its advantages. You added some comments on p4 (line 92) and p20 (line 436). I would say that structure information is missing with monosaccharide assay, while HILIC-FLD provides glycan structures, LC-MS can provide locations of glycosylation. Please consider including this comment in your final manuscripts.

Line 66, for HILIC we edited the following sentence to reflect this comment:

For example, Hydrophilic Interaction Chromatography with fluorescence detection (HILIC-FLD) of N-glycans released with Peptide N-glycosidase F (PNGase F) works well in this context as it allows for the rapid identity of complete glycans when no co-elution occurs.

Line 68, for LC-MS we edited the following sentence to reflect this comment

The identity of glycan isomers by proteolytic digestion and glycopeptide analysis by Liquid Chromatography-Mass Spectrometry (LC-MS) is another powerful alternative and is especially useful to identify the location of the glycans and analyze glycan microheterogeneity on different sites, but requires a larger investment of time and resources to perform.

Line 91, for HPAEC-PAD we edited the following sentence to reflect this comment:

HPAEC-PAD was used in the past to analyze glycoproteins. It waned in popularity due to the development of techniques such as HILIC-FLD which are better suited for identifying the simpler glycan populations found on mAbs. While monosaccharide analysis does not provide glycan identity, it does give information about the average composition of monosaccharide units.

2. On P23, Line 525: "(Agilent, GK80040) at 37°C for 18 h." This is not a complete sentence. It seems that you missed some information for sample preparation for Sialic Acid Analysis.

Thank you for noticing this oversight, we edited line 466 to reflect your comment:

The dried protein sample (25 µg) was treated in triplicate with of 5X Buffer B (10 µl) and Sialidase A (2 µL) (Agilent, GK80040) and adjusted to a final concentration of 0.48µg/µL. The sample was kept at 37°C for 18 h.

3. For Figures 3,4 and 7. You have Y-offsets for overlaid chromatograms. I mean that, using Fig.4 as an example, you have 350 y-offset, the y starts from 0 for the bottom chromatogram, but start from 350 for the overlaid chromatogram. You should add a note in the figure caption.

Thank you for clarifying in regards to the offsets. We clarified in the caption of Figures 3,4, and 7 that y-offsets were added to facilitate comparison.